# Hemp-Derived Nanovesicles Protect Leaky Gut and Liver Injury in Dextran Sodium Sulfate-Induced Colitis

**DOI:** 10.3390/ijms23179955

**Published:** 2022-09-01

**Authors:** Jung-Young Eom, Sang-Hun Choi, Hyun-Jin Kim, Dong-ha Kim, Ju-Hyun Bae, Gi-Seok Kwon, Dong-hee Lee, Jin-Hyeon Hwang, Do-Kyun Kim, Moon-Chang Baek, Young-Eun Cho

**Affiliations:** 1Department of Food and Nutrition, Andong National University, Andong 36729, Korea; 2Department of Molecular Medicine, Cell & Matrix Research Institute, School of Medicine, Kyungpook National University, Daegu 41944, Korea; 3Department of Horticulture & Medicinal Plant, Andong National University, Andong 36729, Korea; 4Industry Academy Cooperation Foundation, Andong National University, Andong 36729, Korea; 5Korea Zoonosis Research Institute, Jeonbuk National University, Iksan 54531, Korea

**Keywords:** inflammatory bowel disease, hemp-derived nanovesicles, oxidative stress proteins, tight junction, gut-liver axis

## Abstract

Hemp (*Cannabis sativa* L.) is used for medicinal purposes owing to its anti-inflammatory and antioxidant activities. We evaluated the protective effect of nanovesicles isolated from hemp plant parts (root, seed, hemp sprout, and leaf) in dextran sulfate sodium (DSS)-induced colitis in mice. The particle sizes of root-derived nanovesicles (RNVs), seed-derived nanovesicles (SNVs), hemp sprout-derived nanovesicles (HSNVs), and leaf-derived nanovesicles (LNVs) were within the range of 100–200 nm as measured by nanoparticle tracking analysis. Acute colitis was induced in C57BL/N mice by 5% DSS in water provided for 7 days. RNVs were administered orally once a day, leading to the recovery of both the small intestine and colon lengths. RNVs, SNVs, and HSNVs restored the tight (ZO-1, claudin-4, occludin) and adherent junctions (E-cadherin and α-tubulin) in DSS-induced small intestine and colon injury. Additionally, RNVs markedly reduced NF-κB activation and oxidative stress proteins in DSS-induced small intestine and colon injury. Tight junction protein expression and epithelial cell permeability were elevated in RNV-, SNV-, and HSNV-treated T84 colon cells exposed to 2% DSS. Interestedly, RNVs, SNVs, HSNVs, and LNVs reduced ALT activity and liver regeneration marker proteins in DSS-induced liver injury. These results showed for the first time that hemp-derived nanovesicles (HNVs) exhibited a protective effect on DSS-induced gut leaky and liver injury through the gut–liver axis by inhibiting oxidative stress marker proteins.

## 1. Introduction

Inflammatory bowel disease (IBD), including ulcerative colitis (UC) and Crohn’s disease (CD), is characterized by acute and chronic inflammatory disorders of the intestine [1,2]. IBD pathogenesis is a prominent feature of intestinal epithelial barrier dysfunction [3]. In IBD, a dysfunctional intestinal epithelial barrier leads to increased gut leakage and induces a host immune response [4]. IBD patients have a dysfunctional intestinal epithelial barrier with increased gut permeability [5]. The dextran sodium sulfate (DSS)-induced colitis mouse model elevates gut leakage by decreasing tight junction (TJ) protein expression, mimicking pathophysiology of IBD in patients. Several recent studies have focused on elevating tight junction protein expression to prevent gut leakage in DSS-induced colitis models [6].

Hemp (*Cannabis sativa* L.) has long been used for therapeutic, spiritual, religious, or recreational purposes. The compounds present in hemp exert antioxidant and anti-inflammatory properties [7] and offer several pharmacological benefits, such as anxiolytic, antipsychotic, anti-rheumatoid arthritis, neuroprotective, and antidiabetic activities [8,9,10]. Additionally, their anti-Alzheimer’s and -Parkinson’s activities have been reported [11]. Recently, several reports demonstrated that IBD patients can benefit from using this plant or its derivatives (cannabidiol) [12,13,14,15,16,17,18]. Therefore, cannabidiol targets enteric reactive gliosis and counteracts the inflammatory environment induced by LPS in mice and in human colonic cultures derived from UC patients [12,13,14,15,16,17,18].

Recently, vegetable- and fruit-derived nanovesicles (NVs) containing proteins, lipids, and RNAs have been identified [19,20], gaining attention as potential therapeutic agents with anticancer [21], anti-angiogenic [22], and anti-inflammatory [23] activities. After oral administration, grape NVs successfully targeted intestinal stem cells and protected mice from dextran sulfate sodium (DSS)-induced colitis [24]. Broccoli nanovesicles also inhibit DSS-induced colitis in mice by activating dendritic cell AMP-associated protein kinase [25]. Therefore, we investigated whether hemp root, seed, sprout, and leaf-derived nanovesicles protected against DSS-induced colitis in mice.

This study showed for the first time isolated and characterized nanovesicles from different plant parts: roots, seeds, hemp sprouts, and leaves. We then investigated whether root-derived nanovesicles (RNVs), seed-derived nanovesicles (SNVs), hemp sprout-derived nanovesicles (HSNVs), and leaf-derived nanovesicles (LNVs) had protective effects against DSS-induced acute colonic and liver injury by inhibiting NF-κB activation and oxidative stress activity.

## 2. Results

### 2.1. RNVs, SNVs, HSNVs, and LNVs Were Purified and Characterized

Nanovesicles were isolated from homogenized hemp roots, seeds, sprouts, and leaves using mammalian extracellular vesicle purification techniques [26]. Root-derived nanovesicles (RNVs), seed-derived nanovesicles (SNVs), hemp sprout-derived nanovesicles (HSNVs), and leaf-derived nanovesicles (LNVs) were characterized by electron microscopic examination (Figure 1A). The size distribution and concentration were measured by using NTA. The sizes and concentrations of RNVs, SNVs, HSNVs, and LNVs were 127 nm and 6.84 × 10^7^ particles/mL, 180 nm and 1.44 × 10^8^ particles/mL, 140 nm and 2.24 × 10^7^ particles/mL, and 107 nm and 6.63 × 10^7^ particles/mL, respectively (Figure 1B). Protein concentrations of RNVs, SNVs, HSNVs, and LNVs were determined by BCA analysis and were 873, 4906, 4205, and 3126 µg in 1.25 g of hemp root, seed, sprout, and leaf, respectively (Figure 1C).

### 2.2. RNVs, SNVs, HSNVs, and LNVs Alleviated Disease Severity and Plasma NO and TNF-α in DSS-Induced Acute Colitis

RNVs, SNVs, HSNVs, and LNVs (1 mg/kg/day) were administered orally for 7 days before DSS treatment to investigate whether they ameliorated DSS-induced colitis in vivo. Hemp-derived nanovesicles’ dose was set to 1 mg/kg/day to mimic the regenerative activity dose of grape nanovesicles [24]. In the DSS-induced mouse model, IBD symptoms, including weight loss and the production of loose, bloody stools, were classified by the disease activity index (DAI) score (Figure 2A). Nitric oxide (NO) is an indicator of oxidative stress in the mucous membrane of IBD patients. Tumor necrosis factor-alpha (TNF-α) is a cytokine marker that activates Th1 cells and deteriorates IBD. RNVs, SNVs, HSNVs, and LNVs treatment significantly reduced the plasma TNF-α levels that were elevated in DSS-induced mice colitis (Figure 2B). In addtion, plasma NO levels were increased in DSS-induced mice colitis, whereas RNVs and SNVs treatment significantly reduced these levels (Figure 2C). These results demonstrated that RNVs and SNVs markedly reduced the inflammatory response and oxidative stress in DSS-induced mice colitis.

### 2.3. RNVs, SNVs, HSNVs, and LNVs Restored TJ/AJ Proteins and Reduced NF-κB Activation and Oxidative Stress Markers in the Colon of DSS-Induced Mice Colitis

DSS-induced colitis in mice is characterized by histological changes, such as crypt abscess, edema, infiltration of inflammatory cells, and destruction of epithelial cells and decreased colon length. Interestingly, NVs, SNVs, HSNVs, and LNVs recovered colon length (Figure 3A). Histological analysis revealed severe chronic inflammation due to crypt abscesses and inflammatory infiltration following DSS-induced colonic injury. However, RNVs, SNVs, and HSNVs treatments restored colitis in the epithelial cells (Figure 3B). An impaired epithelial barrier increases the exposure of immune cells to antigens derived from the gastrointestinal tract and deteriorates IBD with inappropriate immune responses. Accordingly, TJ (claudin4, occludin) and AJ (E-cadherin, α-tubulin) proteins in the epithelial barrier were measured to investigate the effects of RNVs, SNVs, HSNVs, and LNVs in the recovery of the epithelial barrier. The DSS group showed reduced TJ and AJ protein levels, whereas RNV-, SNV-, HSNV-, and LNV-treated mice showed markedly increased TJ and AJ protein levels (Figure 3C). Markedly, RNVs protected the leaky gut by restoring the TJ and AJ proteins.

In line with previous reports regarding NF-κB in IBD [27] and DSS-induced colitis [28], DSS-induced colitis showed that phosphorylation of NF-κB was elevated (Figure 3D). However, RNV, SNV, HSNV, and LNV treatment groups showed markedly reduced levels of phosphorylated(p)-NF-κB and p-IκB (Figure 3D). In addition, RNVs markedly reduced the expression of oxidative stress marker proteins, including COX2, 3-NT, and iNOS that were increased in DSS-induced colitis (Figure 3E). These results indicate that RNVs restored intestinal TJ and AJ proteins with the decreased NF-κB activation, preventing DSS-induced colonic injury.

### 2.4. RNVs, SNVs, HSNVs, and LNVs Restored TJ/AJ Proteins and Attenuated NF-κB Activation and Oxidative Stress Markers in the Small Intestine of DSS-Induced Mice Colitis

In addition to the colon, the effects of RNVs, SNVs, HSNVs, and LNVs on the small intestine were also evaluated. The length of the small intestine was significantly decreased in DSS-induced colonic injury as well. However, RNVs, SNVs, HSNVs, and LNVs recovered the length of the small intestine (Figure 4A). Likewise, H&E staining of DSS-induced mice colitis showed severe crypt abscesses and infiltration of inflammatory cells, whereas RNVs, SNVs, HSNVs, and LNVs significantly restored epithelial cells in the small intestine (Figure 4B). TJ (claudin4, occludin) and AJ (E-cadherin, α-tubulin) proteins in the small intestine were also restored in RNVs, SNVs, HSNVs, and LNVs in DSS-induced colitis (Figure 4C). Importantly, RNVs prevented TJ and AJ protein expression in small intestine in DSS-induced acute colitis.

The phosphorylation of NF-κB in the small intestine was elevated in DSS-induced colitis mice, whereas RNVs, SNVs, HSNVs, and LNVs treatment groups showed markedly reduced levels of p-NF-κB and p-IκB (Figure 4D). In addtion, oxidative strss maker proteins (COX2 and 3-NT) in small intestine were increased in DSS-induced colitis mice. However, RNVs markedly reduced the COX2 and 3-NT protein expression (Figure 4E). That is, as similar with results for DSS-induced colonic injury, RNVs and SNVs restored intestinal TJ and AJ proteins with the decreased NF-κB activation, contributing to the prevention of DSS-induced small intestine injury.

### 2.5. RNVs, SNVs, HSNVs, and LNVs Enhanced Epithelial Barrier Function in T84 Cells

The human colon T84 cell line is derived from intestinal epithelial cells, which forms tight junctions when cultured to 100% confluence. Accordingly, the T84 cell line was employed as an intestinal barrier model [29]. The epithelial cell permeability of T84 colonic cells and the underlying mechanisms were evaluated in the absence or presence of 2% DSS exposure with or without RNVs, SNVs, HSNVs, and LNVs (1 or 10 μM) for 24 h to support our in vivo animal data.

Confocal image analysis showed that 2% DSS treatment disrupted the normal distribution of TG proteins including ZO-1 and occludin in T84 cells (Figure 5A,B). However, changes in confocal images caused by 2% DSS were prevented by treatment with RNVs, SNVs, HSNVs, and LNVs (Figure 5A,B). Furthermore, 2% DSS treatment significantly decreased the rate of trans-epithelial electrical resistance (TEER) (Figure 5C) and elevated the permeation of FITC-D4 (Figure 5D). However, co-treatment with RNVs, SNVs, HSNVs, and LNVs blocked abnormal changes in ZO-1 TJ protein levels and cell permeability in T84 colon cells exposed to 2% DSS (Figure 5). These in vitro results are consistent with the in vivo mouse results. Therefore, we can conclude that RNVs, SNVs, and HSNVs protected the tight junction structure disrupted by DSS, which led to the prevention of DSS-induced gut permeability.

### 2.6. RNVs, SNVs, HSNVs, and LNVs Attenuated the DSS-Induced Acute Liver Injury

Our group recently reported that the DSS-exposed mice modulated liver injury through gut-liver axis [30]. Thus, we evaluated the potential protective role of RNVs, SNVs, HSNVs, and LNVs administration in DSS-induced liver injury. DSS markedly elevated the levels of plasma ALT and AST, while RNVs, SNVs, HSNVs, and LNVs administration reduced the increased ALT and AST levels in DSS-exposed mice (Figure 6A).

H&E-stained histology analysis showed significantly increased hepatocytes damage in DSS-exposed mice, whereas RNVs, SNVs, HSNVs, and LNVs administration blunted these changes (Figure 6B).

The levels of hepatic regeneration Cyclin D 1 and eNOS were decreased by DSS exposure, while the amounts of these proteins were restored by RNVs, SNVs, HSNVs, and LNVs administration (Figure 6C). Our results demonstrated that RNVs, SNVs, HSNVs, and LNVs administration importantly increased hepatic regeneration in DSS-exposed mice.

## 3. Discussion

In this study, the root (RNVS), seed (SNVS), hemp sprout (HSNVs), and leaf (LNVs) of hemp (*Cannabis sativa* L.) were processed to isolate nanovesicles similar in size and structure to those of mammalian-derived extracellular vesicles [26]. Therefore, this is the first study using hemp-derived nanovesicles (RNVs, SNVs, HSNVs, and LNVs) in protection against colitis and liver injury through the gut–liver axis (Figure 7). RNVs significantly restored the TJ/AJ proteins and reduced oxidative stress markers and NF-κB activation in the small intestine and colon in DSS-induced colitis. RNVs, SNVs, and HSNVs restored TJ/AJ proteins and gut permeability levels of DSS-disturbed human colon T84 cells. However, LNVs treatment failed to protect the gut barrier in vitro and in an in vivo model. Interestedly, RNVs, SNVs, HSNVs, and LNVs reduced ALT activity and liver regeneration markers in DSS-induced liver injury.

Recently, several plant-derived nanovesicles (NVs) or exosomes have been reported to attenuate inflammatory bowel disease in a mouse model [24,31,32,33,34]. NVs are inexpensive to make, simple to use, safe, and easily transported into the body. Additional potential benefits of exosomes include a low risk of causing a harmful immune response, a non-neoplastic nature, high stability, and inability to be infected by viruses [35]. The human intestinal mucosal surface provides primary defense against pathogens and regulates the immune response owing to the function of intestinal epithelial cells [36]. The balance between these vesicles in the mucosal microenvironment likely plays an important role in inducing, maintaining, and regulating the necessary functions of intestinal tissue [37]. In addition, plant-derived NVs in which natural phenolic compounds are packaged [38,39] have been shown to have higher stability than vesicles produced by “unnatural” loading of biological components. *Cannabis sativa* L. and its bioactive components have anti-inflammatory, anti-cancer, epilepsy, brain, and skin health effects [38,39,40]. Markedly, oral administration grape NVs protect mice from DSS-induced colitis, and edible plant-derived NVs could be used as therapeutic agents [24]. Therefore, we explored the therapeutic potential of HNVs (RNVs, SNVs, HSNVs, and LNVs) on DSS-induced colitis in mice.

A common feature between IBD and a DSS-induced colitis model is disrupted tight junctions and adherent junction structures [41]. Damage to tight junctions is mediated by the direct effects of DSS [42] or proinflammatory cytokines secondary to inflammation [43]. Barrier dysfunction leads to a pathological condition that induces gut leakage, followed by the migration of symbiotic microorganisms through the body [42,43]. These results in inflammatory and immune responses in tissues other than the gut [42,43]. Our results demonstrated that RNVs restored intestinal TJ and AJ proteins with decreased NF-κB activation, preventing DSS-induced colonic injury.

In our study, RNVs, SNVs, and HSNVs restored the tight junction and adherent junction disrupted by 2% DSS. In an in vitro study using T84 cells exposed to 2% DSS, our data showed an elevation in the expression of tight junction proteins such as ZO-1 and occludin in the presence of RNVs. In addition, DSS-induced tight junction disruption was reversed by RNVs treatment, together with reduced gut permeability in DSS-induced colitis. These results demonstrate that RNVs markedly decreased barrier dysfunction and protected gut permeability from DSS-induced colitis.

NF-κB is a key protein involved in the immune response and plays an important role in the production of cytokines and chemokines in the inflammatory response signaling pathway [44]. When a foreign antigen inflames cells, the IκB kinase (IKK) complex phosphorylates IκB to dissociate NF-κB. Then, NF-κB is transferred into the nucleus, resulting in anti-inflammatory gene expression and activation of immune cells [44]. Reduced phosphorylation of the NF-κB protein and IκBα expression in the colon and small intestine by RNVs, SNVs, and HSNVs implies the anti-inflammatory functions of HNVs.

Numerous studies suggested that IBD is related to damage in multiple organs, including the liver, through the gut–liver axis [45]. Our group reported that FDP exhibits a protective effect on DSS-induced acute and chronic colonic and liver injury through the gut–liver axis via antioxidant and anti-inflammatory properties [30]. Our results showed that liver damage markers such as plasma ALT, AST, and abnormal H&E-stained histology were increased in DSS-exposed mice, whereas RNVs, SNVs, HSNVs, and LNVs administration reduced these liver damage markers. In addition, the expressed levels of hepatic regeneration markers Cyclin D1 and eNOS were significantly reduced in DSS-exposed mice, and these changes were significantly prevented by RNVs, SNVs, HSNVs, and LNVs administration.

Therefore, this is the first study using hemp-derived nanovesicles in protection against DSS-induced colitis and hepatic injury via the gut–liver axis. Hemp-derived nanovesicles contain miRNA, lipids, mRNAs, proteins, and metabolites which have antioxidant and anti-inflammatory effects, although future experiments will be required to confirm this explanation.

## 4. Materials and Methods

### 4.1. Materials

DSS used in this study was obtained from MP Biomedicals (Irvine, CA, USA). The hemp (*Cannabis sativa* L.) used in this experiment was supplied by Gyeongbuk hemp regulation free zone of Andong in November 2020. Other materials not described here were of the highest grade available and reported recently [30,46,47,48].

### 4.2. Nanovesicles of Cannabis sativa L. Isolation

Hemp (*Cannabis sativa* L.) was obtained from Gyeongbuk hemp regulation free zone, Andong, South Korea. The nanovesicles were isolated according to our laboratory methods. Briefly, 1.25 g of washed hemp parts (roots, seeds, sprouts, and leaves) were homogenized separately for 5 min with 10 g of cold phosphate buffer saline (PBS) in a blender to obtain the juice. The juice was then centrifuged twice at 500× *g* for 10 min, 2000× *g* for 20 min, and 10,000× *g* for 30 min to remove the fiber. Each supernatant was further centrifuged at 100,000× *g* for 1 h, and the pellet was resuspended in PBS.

### 4.3. Nanoparticle Tracking Analysis and TEM Analysis

The number and size of hemp-derived nanovesicles (RNVs, SNVs, HSNVs, and LNVs) were measured by nanoparticle tracking analysis (NTA) using a NanoSight NS300 system (NanoSight, Malvern, UK). The instrument was calibrated with 100 nm polystyrene beads (Thermo Fisher Scientific, Waltham, MA, USA) before use. NTA software was used to determine the concentration of nanoparticles (particles/mL). The batch process included in NTA software was used to integrate three measurements of each sample.

For the negative staining and TEM analyses, samples were applied to a glow-discharged copper grid coated with a continuous carbon film and stained with 0.75% uranyl formate as previously described [26].

### 4.4. Animals Models

All animal experimental procedures were performed in accordance with the Andong National University guidelines for small animal experiments and approved by the Andong National University Animal Care and Use Committee (2020-2-0706-01-01). Male C57BL/6 mice (6–8 weeks old) were purchased from Orient Bio Inc. (Seongnam-si, Korea)—and maintained under controlled lighting (12 h light/dark cycle) with ad libitum access to food and water. Acute colitis was induced by adding 5% DSS (MP Biomedicals; Irvine, CA, USA) to the drinking water for 7 days. The animals were randomly assigned to six groups (n = 5–10 mice per group) as follows: (a) water group: fed with standard feed and given free access to drinking water; (b) DSS group: fed with standard feed and given free access to drinking water containing 5% DSS; (c–f) DSS + HNVs group: fed with standard feed and given free access to drinking water containing 5% DSS during the oral administration of (c) RNVs (1 mg/kg/daily), (d) SNVs (1 mg/kg/daily), (e) HSNVs (1 mg/kg/daily), or (f) LNVs (1 mg/kg/daily) for 7 days (n = 5–10/group).

### 4.5. Assessment of Colitis

Colon tissue from the ileocecal junction to the anal verge was washed with PBS to remove fecal residues, blotted on filter paper, weighed, and measured to assess DSS-induced acute colitis score. According to previously reported criteria [30,49], the colons were scored for visible damage on a scale of 0–4.

### 4.6. Plasma TNF-α and NO Analyses

Protein concentration in the samples was measured using a BCA protein assay (Pierce, Rockford, IL, USA). Plasma levels of TNF-α (R&D Systems, Minneapolis, MN, USA) and NO (Cayman, Ann Arbor, MI, USA) were detected using ELISA kits according to the manufacturer’s protocols [30,46,47,48].

### 4.7. Western Bolt Analysis

Tissues were washed twice with PBS and homogenized with 1X RIPA buffer containing protease inhibitor cocktails (Sigma-Aldrich, St. Louis, MO, USA) and a phosphate inhibitor socket (Sigma-Aldrich) to extract proteins from the small intestine and colon tissues of mice treated with RNVs, SNVs, HSNVs, and LNVs for 7 days. Equal amounts of protein were separated by 10% SDS-PAGE and transferred to a nitrocellulose membrane. The membrane was blocked at room temperature for 2 h with a Tris-buffer containing 0.05% Tween 20 (TBS-T) and 5% non-fat dry milk, followed by a reaction at 4 °C for 15 h with the primary antibody. The membrane was washed with TBS-T and incubated with a secondary antibody for 1 h. Protein in the resultant membrane was identified using a chemiluminescence kit (Amersham Biosciences, Piscataway, NJ, USA). Each band was quantified by measuring the intensity using Fusion Solo 6X basic 0.84 (Vilber, Collegien, France). The primary antibodies used were as follows: 3-Nitrutyrisine (1:3000, Cat. No. ab61392, Abcam, Cambridge, UK), COX-2 (1:1000, Cat. No. SC-376861, Santa Cruz Biotechnology Inc., Dallas, TX, USA), p-IκB-α (1:1000, Cat. No. SC-8404, Santa Cruz), p-NF-κB (1:1000, Cat. No. SC-271908, Santa Cruz Biotechnology), occludin (1:1000, Cat. No. SC-271842, Santa Cruz Biotechnology Inc.), E-Cadherin (1:1000, Cat. No. SC-8426, Santa Cruz Biotechnology Inc.), claudin-4 (1:1000, Cat. No. SC-376643, Santa Cruz Biotechnology Inc.), α-tubulin (1:1000, Cat. No. SC-376643, Santa Cruz Biotechnology Inc.), cyclin D1 (1:1000, Cat. No. SC-8396, Santa Cruz Biotechnology Inc.), eNos (1:1000, Cat. No. ab76198, Abcam), and β-actin (1:1000, Cat. No. SC-47778, Santa Cruz Biotechnology Inc.).

### 4.8. Cell Culture

The human colon-cancer-derived cell line, T84, was obtained from American Type Culture Collection (ATCC). The T84 cell line was grown in a culture medium composed of Dulbecco’s modified Eagle’s medium (DMEM)/F12 (DMEM/F-12; HyClone), 10% fetal bovine serum (FBS; Gibco), and 1% penicillin–streptomycin (Thermo Fisher Scientific) at 37 °C and 5% CO_2_. The medium was changed every alternate day. Cells (1 × 10^4^ cells/well in an 8-chamber plate) were cultured in growth media at 5% CO_2_ at 37 °C for 24 h and then treated with HNVs (0, 1, and 10 μg/mL) for 14 d.

### 4.9. Immunofluorescence Staining

T84 was seeded at 1 × 10^5^ cells/well and incubated for 14 days. HNVs (0, 1, and 10 μg/mL) were pretreated to T84 cells for 24 h when their confluence in the well reached 80%. Then, the cells were incubated with 2% DSS in DMEM/F-12 and each HNVs (0, 1, and 10 μg/mL) for 24 h. After treatment, the cells were washed with PBS and treated with 2.5% paraformaldehyde (PFA) for 30 min at room temperature, followed by 0.3% Triton X-100 (Sigma-Aldrich, USA) in PBS for 15 min. The resultant fixed cells were incubated with anti-ZO-1 (Abcam) and anti-occludin (Santa Cruz Biotechnology Inc.) antibodies at 4 °C overnight. Finally, tight junction proteins ZO-1 and occludin in T84 cells were secondary-stained with Alexa Fluor™ 488 goat anti-rabbit IgG (1:300) (Thermo Fisher Scientific), and nuclei were stained with DAPI (Sigma-Aldrich).

### 4.10. Trans-Epithelial Electrical Resistance and FITC-D4 Permeability Analysis

Colonic T84 cells were grown in Transwell inserts with a surface area of 0.33 cm^2^ and 0.4 µm pore size (Thermo Fisher Scientific) [48]. The rates of trans-epithelial electrical resistance (TEER) of monolayer cells were measured with an epithelial volt-ohmmeter in each insert and then multiplied by the membrane surface area (0.33 cm^2^). The results were corrected by subtracting the background resistance of the blank membrane (i.e., no cells). Data were collected from triplicate inserts per treatment in two independent experiments and expressed as the percentage of basal TEER (600 Ω·cm^2^) obtained before treatment. At the end of the TEER measurements, FITC-labeled 4-kDa dextran (1 mg/mL FITC-D4; Sigma-Aldrich) was added to the apical side of the cells. Monolayer permeability was assessed by spectrophotometrically measuring the fluorescence of FITC-D4 in the basal medium compartment using a microplate reader at excitation and emission wavelengths of 485 and 540 nm, respectively. Data are reported as relative fluorescence units (% baseline).

### 4.11. Endotoxin Assay

Plasma endotoxin levels were determined using the commercially available endpoint LAL Chromogenic Endotoxin Quantitation Kit with a concentration range of 0.015–1.2 EU/mL (Thermo Fisher Scientific), as described [30,46,47,48].

### 4.12. Histopathology and Serum ALT Measurement

In this study, in order to observe microscopic changes in intestinal and liver tissues, the small/large intestines and liver of the mice were excised and fixed in 4% neutral formalin, followed by dehydration and paraffin treatment. Paraffin-embedded formalin-fixed blocks of individual liver and ileum were cut to 4 µm thickness and then stained with hematoxylin and eosin (H&E) at the Kyungpook National University (KNU) core lab. The plasma ALT and AST level in each mouse was determined using the standard end-point colorimetric assay kit (Bio Vision, Milpitas, CA, USA), as described [30,46,47,48].

### 4.13. Statistical Analysis

Data were analyzed using SPSS (version 27.0; SPSS Inc., Chicago, IL, USA), and a mean difference of *p* < 0.05 was considered significant. Different letters in the actual figures indicate significant differences between various treatment groups at *p* < 0.05, as determined by one-way ANOVA. Once significant difference was recognized, Tukey’s HSD test as a post hoc analysis was conducted to compare the differences between the different groups. Other methods not described in this study were the same as recently reported [30,46,47,48].

## 5. Conclusions

In summary, we successfully identified and characterized edible-plant nanovesicles from different hemp (*Cannabis sativa* L.) parts (root; RNVS, seed; SNVS, hemp sprout; HSNVs, leaf; LNVs). RNVs markedly alleviated leaky gut and intestinal barrier proteins such as tight junction and adherent junction proteins and reduced NF-κB activation and oxidative stress markers in DSS-induced colitis. Additionally, NVs, SNVs, HSNVs, and LNVs administration reduced liver damage markers and elevated liver regeneration markers. Therefore, this is the first study using hemp-derived nanovesicles in protection against colitis that can be a novel therapeutic strategy to treat IBD.

## Figures and Tables

**Figure 1 ijms-23-09955-f001:**
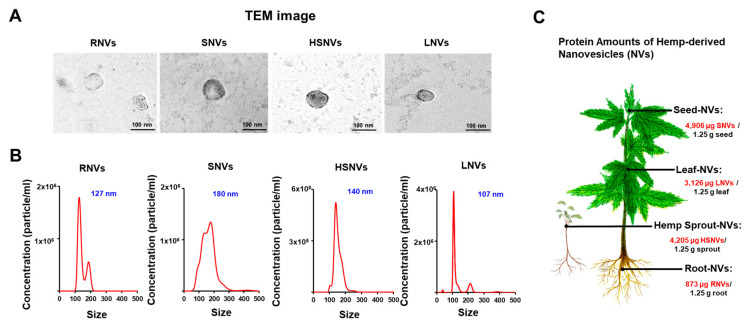
Characterization of RNVs (root-derived nanovesicles), SNVs (seed-derived nanovesicles), HSNVs (hemp sprout-derived nanovesicles), and LNVs (leaf-derived nanovesicles). (**A**) Representative of transmission electron microscope (TEM). (**B**) Nanoparticle tracking analysis (NTA) of size and concentration in the sample of isolated RNVs, SNVs, HSNVs, and LNVs. (**C**) Quantified protein in hemp-derived nanovesicles (HNVs) per 1.25 g. Each part; root-derived nanovesicles (RNVs), seed-derived nanovesicles (SNVs), hemp sprout-derived nanovesicles (HSNVs), leaf-derived nanovesicles (LNVs).

**Figure 2 ijms-23-09955-f002:**
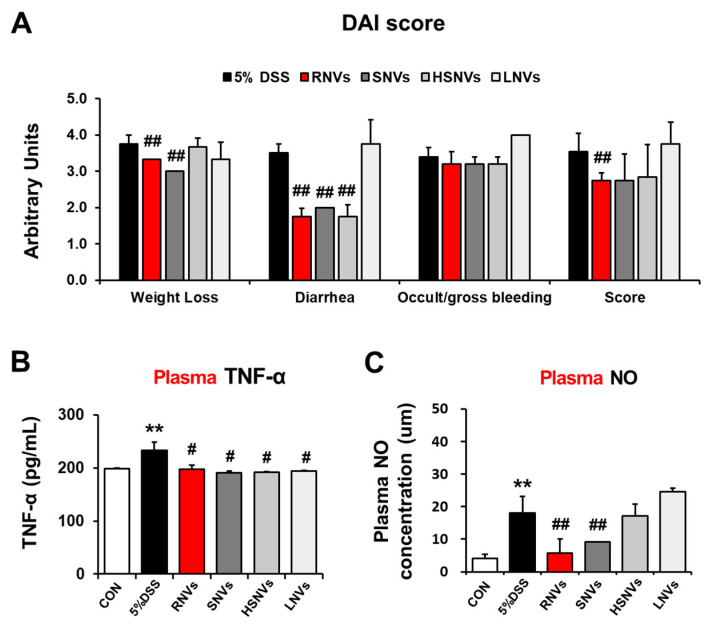
RNVs, SNVs, HSNVs, and LNVs attenuated the disease severity and levels of plasma TNF-α and NO in DSS-induced acute colitis. (**A**) Acute UC was induced by the administration of 5% DSS in drinking water ad libitum for 7 days. Body weights of all mice were measured daily, and daily changes of the average body weight of each group are presented. Total DAI scores with or without RNVs, SNVs, HSNVs, and LNVs administration in DSS-induced acute colitis in mice evaluated at the end of the treatment are shown. (**B**) Plasma levels of TNF-α and (**C**) NO measured by ELISA. ** *p* < 0.01 between 5% DSS and control groups; ## *p* < 0.01. # *p* < 0.05 between DSS vs. RNVs, SNVs, HSNVs, and LNVs groups. Significance of the values for each group was determined using ANOVA and Tukey’s HSD test. Data represent means ± S.E.M. (n = 5–10/group). DAI, disease activity index; NO, nitric oxide; TNF-α, tumor necrosis factor α.

**Figure 3 ijms-23-09955-f003:**
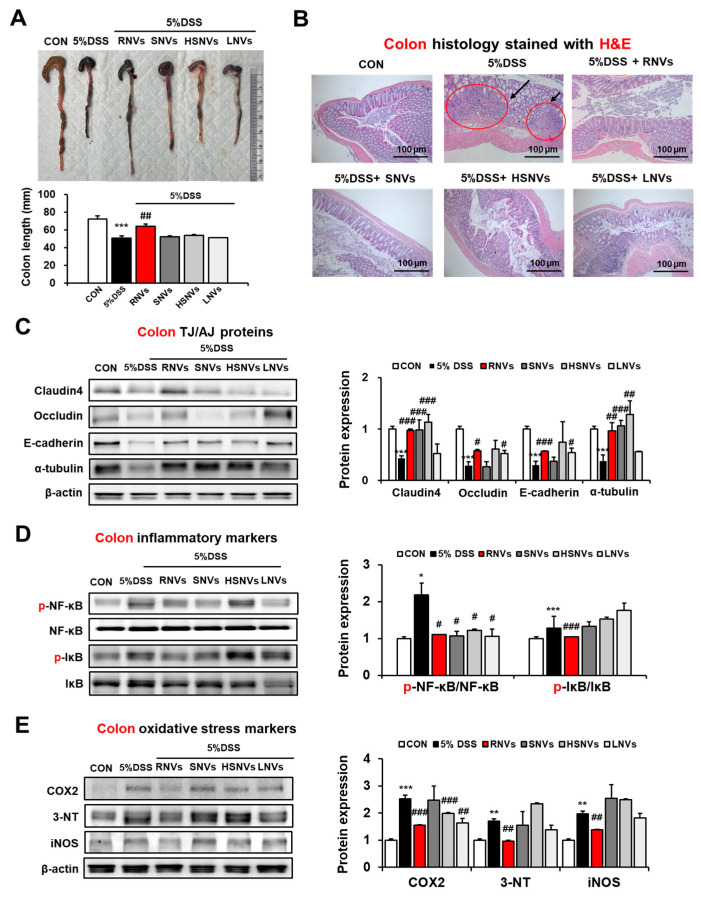
RNVs, SNVs, HSNVs, and LNVs restored the levels of TJ/AJ proteins and attenuated NF-κB activation and oxidative stress markers in the colon of mice with DSS-induced colitis. (**A**) Comparison of colon lengths. (**B**) Representative H&E stain of formalin-fixed colon sections for control (CON), 5% dextran sulfate sodium (5% DSS), and 5% DSS+ RNVs, SNVs, HSNVs, and LNVs-exposed mice. (**C**) The levels of TJ protein (claudin-4, occludin) and AJ protein (E-cadherin, α-tubulin) in the indicated groups are presented. (**D**,**E**) The levels of inflammatory marker proteins (p-NF-κB, p-IκB) and oxidative stress marker proteins (COX2, 3-NT, iNOS) in the indicated groups are presented. Densitometric analysis of immunoblots for each protein relative to β-actin is shown. * *p* < 0.05, ** *p* < 0.01, *** *p* < 0.001 between 5% DSS and control groups; # *p* < 0.05, ## *p* < 0.01, ### *p* < 0.001 between 5% DSS vs. RNVs, SNVs, HSNVs, and LNVs groups. Significance of mean values for each group was determined using ANOVA and Tukey’s HSD test. Data represent means ± S.E.M. (n = 5–10/group).

**Figure 4 ijms-23-09955-f004:**
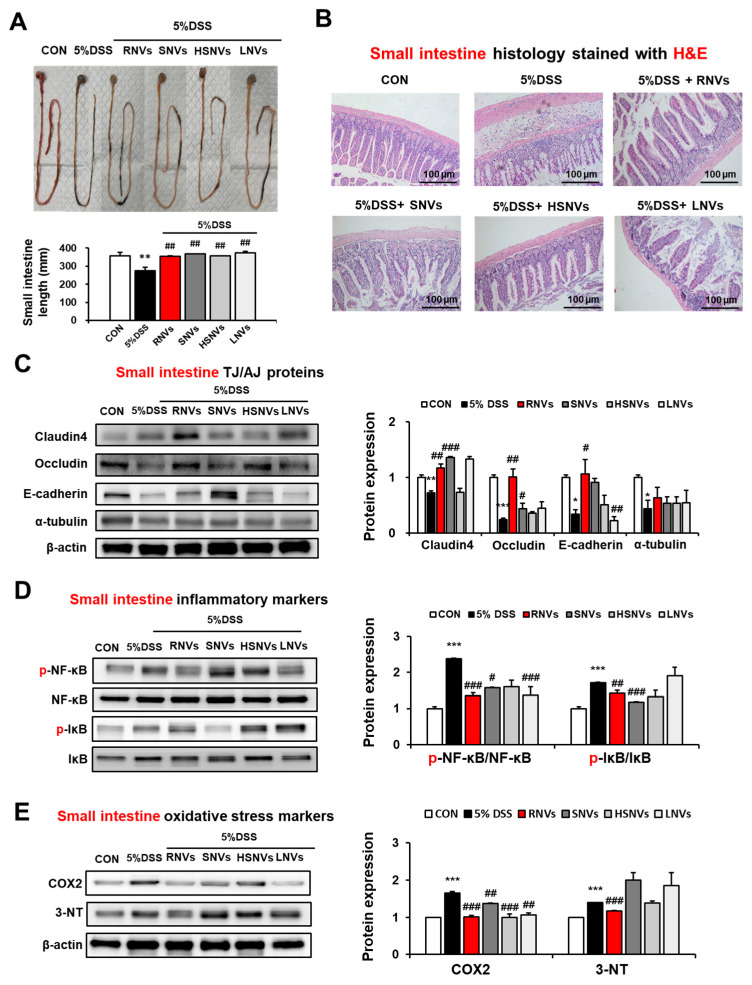
RNVs, SNVs, HSNVs, and LNVs restored the levels of TJ/AJ proteins and attenuated NF-κB activation and oxidative stress markers in the small intestine of mice with DSS-induced colitis. (**A**) Comparison of small intestine lengths. (**B**) Representative H&E of formalin-fixed small intestine sections for control (CON), 5% DSS, and 5% DSS+ RNVs, SNVs, HSNVs, and LNVs-exposed mice. (**C**) The levels of TJ protein (claudin-4, occludin) and AJ protein (E-cadherin, α-tubulin) in the indicated groups are presented. (**D**,**E**) The levels of inflammatory marker proteins (p-NF-κB, p-IκB) and oxidative stress marker proteins (COX2, 3-NT) in the indicated groups are presented. Densitometric evaluation of immunoblots for each protein relative to β-actin is shown. * *p* < 0.05, ** *p* < 0.01, *** *p* < 0.001 between 5% DSS and control groups; # *p* < 0.05, ## *p* < 0.01, ### *p* < 0.001 between DSS vs. RNVs, SNVs, HSNVs, and LNVs groups. Significance of the mean values for each group was determined using ANOVA and Tukey’s HSD test. Data represent means ± S.E.M. (n = 5–10/group).

**Figure 5 ijms-23-09955-f005:**
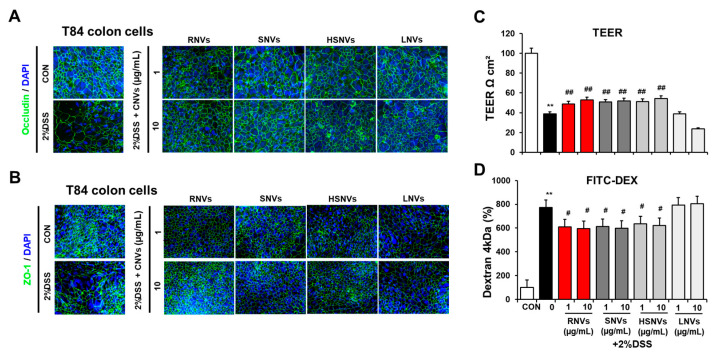
RNVs, SNVs, HSNVs, and LNVs protected the levels of TJ proteins in DSS-induced human colon-cancer-derived cell line T84. (**A**,**B**) Immunofluorescence stain by ZO-1 and occludin of T84 cells for sections for control (CON), 5% DSS, and 5% DSS+ RNVs, SNVs, HSNVs, and LNVs-exposed. (**C**,**D**) Representative levels of TEER and permeability to FITC-D4 after pretreatment without or with the RNVs, SNVs, HSNVs, and LNVs. Data indicate means ± SD of triplicate wells from two separate experiments. ** *p* < 0.01 between 5% DSS and control groups; # *p* < 0.05, ## *p* < 0.01 between DSS vs. RNVs, SNVs, HSNVs, and LNVs groups. Significance of the mean values for each group was determined using ANOVA and Tukey’s HSD test.

**Figure 6 ijms-23-09955-f006:**
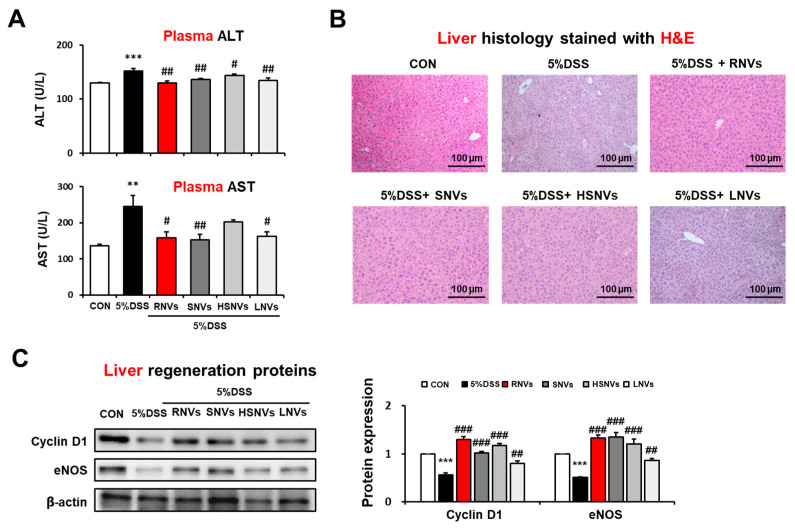
RNVs, SNVs, HSNVs, and LNVs attenuated liver injury in DSS-induced acute colitis mice. (**A**) Levels of plasma ALT/AST and (**B**) H&E-stained sections of liver are shown. (**C**) The levels of liver regeneration markers Cyclin D1 and eNOS in the indicated groups are presented. Densitometric evaluation of immunoblots for each protein relative to β-actin is shown. ** *p* < 0.01, *** *p* < 0.001 between 5% DSS and control groups; # *p* < 0.05, ## *p* < 0.01, ### *p* < 0.001 between DSS vs. RNVs, SNVs, HSNVs, and LNVs groups. Significance of the mean values for each group was determined using ANOVA and Tukey’s HSD test. Data represent means ± S.E.M. (n = 5–10/group).

**Figure 7 ijms-23-09955-f007:**
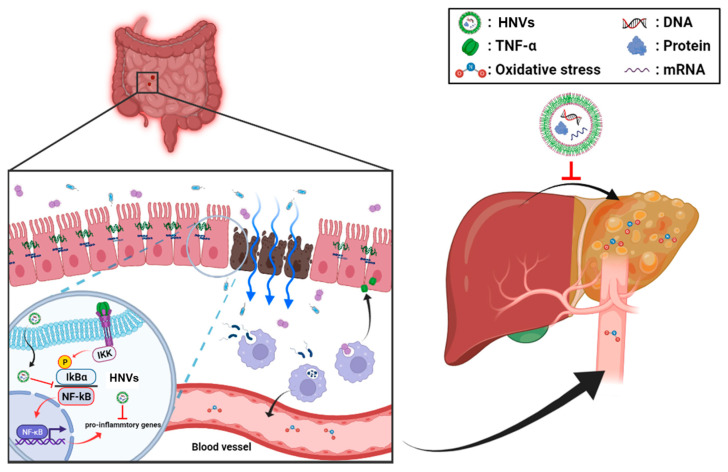
Summary of the preventive effects of hemp-derived nanovesicles against DSS-induced acute colitis, leaky gut, and liver injury through the gut–liver axis.

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
