# Peer review of "Hemp-Derived Nanovesicles Protect Leaky Gut and Liver Injury in Dextran Sodium Sulfate-Induced Colitis"

_ijms, 2022, doi:10.3390/ijms23179955_

Round 1
Reviewer 1 Report
I have evaluated the manuscript with interest. I appreciated the experiments you have done, which provided theoretical support for the development of Hemp (Cannabis sativa L.) as a treatment for colitis.
There are several minor criticisms as follow:
1. Materials and Methods section, the source and purity of the major materials and reagents used need to be described.
2. Animal models
In most acute colitis models with DSS, the dose is around 3%, why use such a high dose (5%)?
How the gavage dose is selected (1 mg/kg/daily)?
3. In animal and cell experimental protocols, more details need to be paid attention to. Are nanovesicles treatment preventive, interventional or therapeutic?
4. Are the nanovesicles concentrations used in animal (1 mg/kg/daily) and cell (1 and 10 ug/ml) experiments prepared from lyophilized powders from supernatant?
5. How toxic are the nanovesicles to cells?
6. Section 4.1, How much washed hemp parts sample were homogenized?
7. What are the main components of Nanovesicles? What substances play an antioxidant and anti-inflammatory role?
8. DSS induces colitis. Why does the small intestine show severe inflammation?
9. Lines 224, missing a eNOS
Reviewer 2 Report
General
Manuscript by Eom et al. is a well-written manuscript which covers an up-to-date topic regarding a probable protective effect of hemp in colitis. Generally results are comprehensive and I have no objections to this part. However, I have some minor suggestions and remarks related to Introduction, Materials and Methods and Discussion which are listed below:
Introduction
-I would suggest to clarify the first sentence and to indicate that while Crohn’s disease may affect any part of the gastrointestinal tract, ulcerative colitis is limited only to colon or rectum.
-Line 55 – what are the benefits of using hemp for IBD patients? Authors must explain it in more details.
-Authors write that currently nanovesicles are gaining more attention. However, it is unknown whether this is the first study examining a protective influence of hemp-derived nanovesicles on DSS-induced colitis? If there are more studies it would be worth citing them and briefly describe them.
Materials and Methods
-What was the purpose of allocating different number of mice in the following study groups?
-Please give a rationale for using the amount of hemp-derived nanovesicles of 1 mg/kg/daily. Are there any studies in which such amount of hemp was justified?
-Line 395 – I would suggest to mention the names of statistical tests used in the study.
Discussion
-Line 259 – It would worth highlighting if this is the first study using hemp-derived nanovesicles in protection against colitis.
-Please provide strengths and limitations of the present study.
Author Response
Introduction
-I would suggest to clarify the first sentence and to indicate that while Crohn’s disease may affect any part of the gastrointestinal tract, ulcerative colitis is limited only to colon or rectum.
Response: We appreciated the reviewer's comment. To respond to this reviewer’s comment, we modified the first sentence.
-Line 55 – what are the benefits of using hemp for IBD patients? Authors must explain it in more details.
Response: We appreciated the reviewer's comment. Therefore, we modified the Introduction (Line 56).
-Authors write that currently nanovesicles are gaining more attention. However, it is unknown whether this is the first study examining a protective influence of hemp-derived nanovesicles on DSS-induced colitis? If there are more studies it would be worth citing them and briefly describe them.
Response: We appreciated the reviewer's comment. Therefore, we added this sentence in Abstract and Introduction (Line 67).
Materials and Methods
-What was the purpose of allocating different number of mice in the following study groups?
Response: We appreciated the reviewer's comment. Animals were randomly assigned to six groups (n = 5–10 mice per group). Since severe 5% DSS treated mice are dead. Therefore, we assigned 10 mice in % DSS treated mouse groups. Also, we modified the method and figure legends.
-Please give a rationale for using the amount of hemp-derived nanovesicles of 1 mg/kg/daily. Are there any studies in which such amount of hemp was justified?
Response: We appreciated the reviewer's comment. Hemp-derived nanovesicles dose was set to 1 mg/kg/day to mimic the regenerative activity dose of grape nanovesicles [1]. Therefore, we added the results (Line 101).
-Line 395 – I would suggest to mention the names of statistical tests used in the study.
Response: We appreciated the reviewer's comment. Therefore, we modified the statistical method (Line 449).
Discussion
-Line 259 – It would worth highlighting if this is the first study using hemp-derived nanovesicles in protection against colitis.
Response: We appreciated the reviewer's comment. To respond to this reviewer’s comment, we added this sentence.
-Please provide strengths and limitations of the present study
Response: We appreciated the reviewer's comment. To respond to this reviewer’s comment, we added the discussion part (Line 319).
References:
- S. Ju, J. Mu, T. Dokland, X. Zhuang, Q. Wang, H. Jiang, X. Xiang, Z.B. Deng, B. Wang, L. Zhang, M. Roth, R. Welti, J. Mobley, Y. Jun, D. Miller, H.G. Zhang, Grape exosome-like nanoparticles induce intestinal stem cells and protect mice from DSS-induced colitis. Mol Ther. 2013;21(7):1345-57.
Round 2
Reviewer 1 Report
This manuscript can be accepted
Reviewer 2 Report
Authors have responded comprehensively to all my suggestions.